# A Real-Time Path Planning Method for Urban Low-Altitude Logistics UAVs

**DOI:** 10.3390/s23177472

**Published:** 2023-08-28

**Authors:** Minyi Deng, Qingqing Yang, Yi Peng

**Affiliations:** Faculty of Information Engineering and Automation, Kunming University of Science and Technology, Kunming 650500, China

**Keywords:** path planning, low-altitude logistics, real-time, UAV

## Abstract

To solve the problem of poor real-time performance in path planning algorithms for unmanned aerial vehicles (UAVs) in low-altitude urban logistics, a path planning method combining modified Beetle Antennae Search (BAS) with the Simulated Annealing (SA) algorithm is proposed. Firstly, based on the requirements of task execution and constraints of UAV flight, a fitness function for real-time search of waypoints is designed while ensuring the safety and obstacle avoidance of the UAV. Then, to improve the search accuracy and real-time performance, determining the initial search direction in the BAS algorithm is improved, while the search step size and antennae sensing length are updated in real-time according to the distance between the UAV and the obstacle. Finally, the SA algorithm is combined with the BAS algorithm to update the waypoints, expanding the search range of each waypoint, avoiding the process of updating the waypoints from becoming trapped in the local optimal waypoints. Meanwhile, the effectiveness of the next waypoint is evaluated based on the Metropolis criterion. This paper generates a virtual urban logistics distribution environment based on the density and distribution of urban buildings, and compares the performance of algorithms in obstacle-sparse, obstacle-moderate, and obstacle-dense environments. The simulation results demonstrate that the improved method in this paper has a more significant capacity for environmental adaptation. In terms of the path length, waypoints, safety obstacle avoidance, and smoothness, the planned path outperforms the original BAS method. It satisfies the needs of real-time path planning for UAVs involved in urban low-altitude logistics.

## 1. Introduce

With the rapid development of e-commerce, the demand for logistics and the pressure of urban ground transportation is constantly increasing. Therefore, the efficiency of logistics distribution has begun to attract attention [1,2]. UAVs, known for their high speed, strong maneuverability, and wide delivery range [3,4,5], have been applied in the delivery of medical supplies, mail, food, and more, becoming one of the tools for urban logistics distribution [6]. Logistics UAVs have been supported and practiced globally [7]. As the world’s largest user of UAVs, China has approved several express delivery companies to conduct pilot UAV delivery services [8]. The sustainable development of aviation transportation is continuously advancing [9]. Therefore, it is crucial to address the challenge of enabling UAVs to complete delivery tasks safely and efficiently in urban low-altitude environments. Among the challenges, the key to solving the problem lies in planning a flight path, which ensures both safety by avoiding obstacles and a relatively short path length.

Thus, the logistics UAV path planning methods have attracted many scholars’ attention. In reference [10], an improved A* algorithm was used to quickly plan feasible paths for urban logistics UAVs, while considering the delivery efficiency and costs. Ref. [11] proposed an algorithm based on the combination of the ant colony algorithm and unnecessary point deletion strategy, which solves the problem of logistics UAV path planning under multiple obstacle conditions. Ref. [12] introduced a phased path planning strategy: in the first phase, routes were computed offline based on static risk factor information, and in the second phase, a Borderland algorithm was presented to dynamically adjust paths using real-time map data. Ref. [13] introduced an improved method for constructing a grid-based aerial environment and a path planning model for trans-oceanic logistics UAVs. It employs the A* algorithm with an enhanced heuristic function to address the path planning issue. The studies mentioned above have conducted in-depth research on path planning for logistics UAVs. However, the complexity of these path planning methods is high, and they are offline, leading to issues such as long planning times, limited autonomy, and weak adaptability to the environment. Real-time path planning allows UAVs to autonomously adjust their paths based on changes in the environment and tasks, enhancing the effectiveness and safety of UAV flights.

Current research on path planning algorithms is mainly categorized into intelligent biomimetic, learning, and search algorithms [14,15]. Among these algorithms, the BAS algorithm [16], as the most representative search algorithm, requires only one individual and is not limited by initial conditions. It has fewer constraints and exhibits lower complexity and stronger search capabilities. Consequently, it holds an advantage in addressing real-time UAV path planning problems. Due to its robust search capabilities and lower complexity, it has been widely adopted by many researchers to solve problems related to UAV path planning. Ref. [17] proposed the path planning method combining ant colony algorithm and BAS algorithm. This method employs targeted search mechanisms to ensure the stability of path finding, avoiding blind exploration and reducing the waste of computational resources. Ref. [18] addressed the contradiction between the high computational complexity of bio-inspired algorithms and real-time UAV path planning. They introduced an obstacle-avoiding BAS algorithm, which generates paths that meet both safety flight requirements and enhance path planning efficiency. Ref. [19] employed the BAS algorithm to select the most suitable gain coefficient for the artificial potential field algorithm, enabling the UAV to avoid obstacles and accurately reach the target position. Ref. [20] proposed an evolutionary algorithm that combines the BAS algorithm and Particle Swarm algorithm, to address the issue of multi-objective optimization path planning. This algorithm utilizes the BAS algorithm to update particles, leading to a reduction in the number of iterations and an enhancement of the search speed. In reference [21], a particle swarm optimization BAS algorithm was utilized to plan a three-dimensional path within the low-altitude airspace. This method effectively reduced the path redundancy and utilized B-spline curves to smoothen the discretely planned path.

The research mentioned above has improved the BAS algorithm, utilizing its powerful search capability to find paths. However, the above studies do not consider that it is the low computational effort of the algorithms that makes path planning simple and easy to fall into local optimality. Especially in low-altitude urban environments, the dense distribution of city buildings increases the complexity of flight path planning [22], raising the probability of falling into local optima. The SA algorithm [23], by randomly generating multiple neighboring solutions and selecting the one that optimizes the fitness function, not only expands the search range and enhances search efficiency but also effectively avoids local convergence. Ref. [24] considers the limited efforts required by other methods, which tend to result in local optima, and proposes an improved dynamic path planning SA algorithm. This algorithm introduces an initial path selection method and deletion operation to make computations faster and more efficient. Ref. [25], based on the particle swarm optimization algorithm, introduces annealing mechanisms and the Metropolis criterion, updating particle states through an SA probability jumping strategy, reducing the probability of particles falling into local optima. In reference [26], the SA algorithm is incorporated into the ant colony algorithm to address the premature convergence in the ant colony algorithm. The traditional ant colony algorithm’s update mechanism is modified, resulting in a more robust and efficient algorithm to complete tasks effectively. These studies indicate that the SA algorithm holds a significant advantage in addressing the problem of other algorithms easily becoming trapped in local optima.

Therefore, a path planning method is proposed, named the Beetle Antennae Search Simulated Annealing (SABAS) algorithm, in response to real-time path planning for urban low-altitude logistics UAVs. This algorithm improves based on the BAS algorithm, which has low complexity and high search capability, and combines with the SA algorithm which can expand the search range to avoid becoming trapped in local optima.

The main contributions of this paper can be summarized as follows:

(1) Considering the constraints during UAV flight, design a fitness function that satisfies safe obstacle avoidance, shortest path, and real-time planning requirements.

(2) Improving the BAS algorithm. To enhance search safety and real-time performance, the search step size and antennal sensing length are dynamically updated based on the distance between the UAV and obstacles. When the UAV is far from obstacles, larger step sizes and antennal sensing lengths are used for searching. In comparison, smaller step sizes and antennal sensing lengths are employed when the UAV is closer to obstacles.

(3) Applying the SA algorithm to the position updating of the BAS algorithm. The search range is expanded by generating neighboring solutions for the next position, preventing the position update process from falling into local optima. Additionally, assessing whether the next position meets the movement criteria based on the Metropolis criterion, to enhances the accuracy of the search. 

(4) Comparing with the BAS algorithm in three urban low-altitude scenarios, where the only difference among these scenarios is the obstacle density and distribution. Test the algorithm’s performance in terms of safe obstacle avoidance, number of waypoints, and flight path length. 

## 2. Algorithm Design

### 2.1. Algorithm Principle

The BAS algorithm is a metaheuristic algorithm that simulates the behavior of beetles searching for mates or prey [16]. When beetles hunt for prey or search for potential mates, they swing their body’s two antennae to detect the scent of prey or potential mates, indicating that beetles use their two antennae to search the surrounding area randomly. When the concentration of scent detected by one side of the beetle’s antenna is higher than that detected by the other side, the beetle will move to the side with the higher concentration. In the BAS algorithm, the fitness function represents the concentration of the scent, and the beetle’s position represents the solution to the problem. The fitness function’s optimal value corresponds to the odor’s source point, and the optimal solution corresponds to the beetle’s optimal position. Because the algorithm’s computation process is relatively simple, initial conditions do not constrain it, and it has fewer constraints. Therefore, this algorithm is widely used in fields such as image processing and intelligent control. However, the search strategy employed by the algorithm is local search, which can lead to the algorithm becoming trapped in optimal local solutions. The SA algorithm expands the search range, improves the search efficiency, and effectively avoids local convergence by randomly generating multiple neighborhood solutions and selecting the solution that makes the fitness function optimal. Therefore, for the problem of poor real-time path planning algorithm, this paper proposes an improved SABAS algorithm, which can meet the demand of real-time obstacle avoidance, and searching of waypoints for urban low-altitude logistics UAVs. 

### 2.2. Algorithm Flow 

The steps to improve the SABAS are as follows:

Step 1 Initialization. Initialize the variables, determine the starting and ending positions, and read the simulated map data.

Step 2 Design the fitness function. According to the specific requirements of UAV path planning, to ensure the safe flight of UAV, design the fitness function of the BAS algorithm to update the waypoints during the path planning process.

Step 3 Initialize the search direction. Initialize the search direction of the beetle according to the starting and ending positions.

Step 4 Calculate the left and right antennae positions. Calculate the beetle’s left and right antennae positions based on the current position and the sensing length of antennae.

Step 5 Update beetle position. Update the next position according to the current position and the position of the left and right antennae, and determine whether the beetle reaches the endpoint. If it reaches the endpoint, go to step 9, otherwise, go to step 6.

Step 6 Expand the search range. The SA algorithm is introduced to generate the neighborhood solutions of the next position obtained from Step 5, and the best next position is selected from the neighborhood solutions.

Step 7 Move evaluation. If the filtering in Step 6 meets the move condition, move to the next step and go to Step 8; if the move condition is not met, go back to Step 6.

Step 8 Update the step size and the sensing length of antennae. According to the distance between the UAV and the obstacle, update the search’s step size and the sensing length of antennae.

Step 9 Obtain the path.

### 2.3. Fitness Function Design


(1)
$$F(X^{t})=c_{1}q^{t}+c_{2}h^{t}+c_{3}l^{t}\;$$


In the formula
(2)$$q^{t}=\left\{ \begin{array}{l} d_{\min}\cos\left(\frac{\pi }{3d_{\min}}d_{t,b}\right),0\leq d_{t,b}\leq d_{\min}\\ \frac{\sqrt{3}}{3}d_{\min}\cos\left(\frac{\pi }{6d_{\min}}d_{t,b}\right),d_{\min}<d_{t,b}\leq 3d_{\min}\\ 0,\;d_{t,b}>3d_{\min} \end{array}\right.\;$$
(3)$$d_{t,b}=\sqrt{{\left(x^{t}-x^{b}\right)}^{2}+{\left(y^{t}-y^{b}\right)}^{2}+{\left(z^{t}-z^{b}\right)}^{2}}$$
(4)$$h^{t}=\frac{z^{t,t-1}}{2}\left(1+\cos\left(\frac{2\pi \left(z^{t}-H_{\min}\right)}{\left(H_{\max}-H_{\min}\right)}\right)\right)$$
(5)$$z^{t,t-1}=\left|z^{t}-z^{t-1}\right|$$
(6)$$l^{t}=\sqrt{{\left(x^{T}-x^{t}\right)}^{2}+{\left(y^{T}-y^{t}\right)}^{2}+{\left(z^{T}-z^{t}\right)}^{2}}$$

The constraints are:(7)$$High_{\min}\leq z^{t}\leq High_{\max}$$
(8)$$0\leq \beta_{t}=\arctan\frac{\left|x^{t+1}-x^{t}\right|}{\left|y^{t+1}-y^{t}\right|}\leq \beta_{\max}$$
(9)$$0\leq \mu_{t}=\arctan\left(\frac{\left|x^{t+1}-x^{t}\right|}{\sqrt{{\left(x^{t+1}-x^{t}\right)}^{2}+{\left(y^{t+1}-y^{t}\right)}^{2}}}\right)\leq \mu_{\max}$$
(10)$$d_{t,b}\geq d_{\min}$$
where, $q^{t}$, $h^{t}$, and $l^{t}$ are respectively the danger cost, altitude cost, and distance cost from the endpoint for the UAV when it is at position $X^{t}$, $High_{\max}$ is the maximum flight altitude, $High_{\min}$ is the minimum flight altitude, $\beta_{\max}$ is the maximum turning angle, $\mu_{\max}$ is the maximum pitch angle, $c_{1}$, $c_{2}$, and $c_{3}$ are the danger cost weight, altitude cost weight, and distance cost weight between the current position and the endpoint, respectively, and $c_{1}$, $c_{2}$, and $c_{3}$ satisfies $c_{1}+c_{2}+c_{3}=1$; $d_{t,b}$ is the minimum safe distance between the UAV and the obstacle, $d_{\min}$ is the distance between the UAV and the obstacle detected by the sensor when the UAV is in position $X^{t}$, $z^{t,t-1}$ and $d_{t,t-1}$ are the height difference and distance between waypoints $X^{t}$ and $X^{t-1}$, $\left(x^{t},y^{t},z^{t}\right)$ is the coordinates of the waypoint $X^{t}$, $\left(x^{t+1},y^{t+1},z^{t+1}\right)$ is the coordinates of the waypoint $X^{t+1}$,$\left(x^{b},y^{b},z^{b}\right)$ is the coordinates of obstacles, $\left(x^{T},y^{T},z^{T}\right)$ is the coordinates of the endpoint and $X^{T}$ of the UAV. Equations (7)–(10) are the flight height constraint, steering angle constraint, pitch angle constraint and the minimum safe distance from the obstacle respectively.

### 2.4. Search Direction

The search direction is a crucial factor affecting the accuracy of beetle position updates. In the original BAS algorithm, the initial search direction is randomly generated using Equation (11), resulting in a relatively arbitrary search direction. The ideal flight path of the UAV is a straight line path from the starting point to the endpoint, so the ideal search direction is the direction from the starting point to the endpoint. The search direction obtained through Equation (11) could be opposite or perpendicular to that direction. Searching in these initial directions would cause the UAV to deviate from the ideal flight path right from the start. Subsequent positions updated in this search direction would also increasingly deviate from the ideal path, leading to increased path length and time, resulting in significant resource wastage. Therefore, to improve the search’s accuracy and shorten the flight’s path length and time of the flight, the search direction is set as the direction from the starting point to the endpoint, and the calculation formula is shown in Equation (12). The beetle moves towards the destination right from the first step. Although it may need to deviate from the ideal path direction due to obstacles along the way, once the beetle bypasses the obstacle, it continues to approach the ideal path direction. This method avoids unnecessary increases in path length and enhances search efficiency.
(11)$$\vec{b}=\frac{\mathrm{rand}(k,1)}{\left||rand(k,1)|\right|}\;$$
(12)$$\vec{b}=\frac{\vec{X^{0}X^{T}}}{\left||X^{T}-X^{0}|\right|}=\frac{\vec{X^{0}X^{T}}}{\sqrt{{\left(x^{T}-x^{0}\right)}^{2}+{\left(y^{T}-y^{0}\right)}^{2}+{\left(z^{T}-z^{0}\right)}^{2}}}\;$$
where, $\vec{b}$ is the search direction, *k* is the spatial dimension, the spatial dimension considered in this paper is three dimensions, $X^{0}$ is the starting position and its coordinates are $\left(x^{0},y^{0},z^{0}\right)$, $X^{T}$ is the end position and its coordinates are $\left(x^{T},y^{T},z^{T}\right)$.

### 2.5. Update Position

Beetles update their position based on the concentration difference between the odors detected by their two antennae. Therefore, the position of the beetle’s left and right antennae must be obtained first, and the calculation formula is given in Equation (13). The formula for updating the beetle’s position is given in Equation (14):(13)$$\begin{array}{l} X_{r}^{t}=X^{t}+d^{t}/2\vec{b}\\ X_{l}^{t}=X^{t}-d^{t}/2\vec{b} \end{array}$$
(14)$$X^{t+1}=X^{t}+\delta^{t}\vec{b}sign\left[F(X_{r}^{t})-F(X_{l}^{t})\right]$$
where, $d^{t}$, $\delta^{t}$, $X_{r}^{t}$ and $X_{l}^{t}$ are the lengths of the antennae sensing, search step size, right antenna position, and left antenna position of the beetle at position $X^{t}$, respectively, $F(X_{r}^{t})$ and $F(X_{l}^{t})$ represent the fitness values of the right and left antennae of the beetle at position $X^{t}$, respectively, $\vec{b}$ is the search direction, and sign(•) is a sign function.

The original BAS algorithm consists of Equations (13) and (14) to update the beetle’s next position, which compares the three forward directions of left front, right front, and right front. It selects the optimal direction to move forward by one step to obtain the position of the next move. The updated position is only obtained by adding one step along the better direction among the three directions, and there is no guarantee that this position is the best moving position. Such a way of updating positions has significant limitations and low accuracy, leading to inaccurate position updates or choosing an unfavorable path to reach the goal.

The SA algorithm can expand the search range by generating neighboring solutions to avoid becoming trapped in local optima. The algorithm uses the Metropolis criterion to handle the obtained solutions, accepting better and worse solutions with a certain probability. Therefore, to compensate for the shortcomings of the BAS algorithm, the SA algorithm is combined with the BAS algorithm to update the position. During the position updating, neighborhood search is used to screen the best moving position while optimizing the moving judgment criteria.

#### 2.5.1. Neighborhood Search Optimization

In this paper, we use the SA algorithm to generate neighborhood solutions to expand the range of the BAS, i.e., after obtaining the next position $X^{t+1}$ to be moved according to Equation (14), we introduce the SA algorithm to generate *M* neighborhood solutions $\{X_{m}^{t+1}|1\leq m\leq M\}$. To obtain a better moving position, obtain ball $O_{M}$ using Equation (15), and $X_{m}^{t+1}$ is taken on the hemisphere boundary of sphere $O_{M}$ in the $\vec{X^{t}X^{t+1}}$ direction. Then, the best next moving position is screened by comparing the difference in the fitness function between the neighborhood solution $X_{m}^{t+1}$ and the initial solution $X^{t+1}$ of the position to be moved, and the screening position is calculated by Equations (16) and (17).
(15)$${\left(x^{t+1}-x^{t}\right)}^{2}+{\left(y^{t+1}-y^{t}\right)}^{2}+{\left(z^{t+1}-z^{t}\right)}^{2}={\left|\vec{X^{t}X^{t+1}}\right|}^{2}$$
(16)$$\Delta F=F(X_{m}^{t+1})-F(X^{t+1})\;$$
(17)$$X^{t+1}=\left\{ \begin{array}{l} X_{m}^{t+1},\\ X^{t+1}, \end{array}\right.\begin{array}{l} \left(\Delta F<0\right)\\ \left(\Delta F\geq 0\right) \end{array}\}\;$$
where $\;X^{t+1}$ is the next position to be moved by Equation (14), $\left(x^{t},y^{t},z^{t}\right)$ is the coordinates of the position $X^{t}$, $\left(x^{t+1},y^{t+1},z^{t+1}\right)$ is the coordinates of the position $X^{t+1}$, $X_{m}^{t+1}$ is the mth neighborhood solution of $\;X^{t+1}$, ΔF  is the difference between the fitness function of the neighborhood solution and the position to be moved, ΔF <0 indicates that the neighborhood solution is better, The neighborhood solution is replaced with the position to be moved. Otherwise, the original position is kept.

#### 2.5.2. Optimization of Moving Conditions

To ensure the effectiveness of beetle position updates, an evaluation of the upcoming position that the beetle is about to move to is required before the beetle’s movement. If the upcoming position to move to is better than the current position or meets certain conditions, the beetle will move to that position. Otherwise, it is necessary to find the optimal moving position by increasing the neighboring solutions. Therefore, this paper assesses whether the next position meets the moving criteria according to the Metropolis criterion. Let the difference in fitness function between the upcoming moving position and the current position be denoted as $\Delta F^{\prime }=F(X^{t+1})-F(X^{t})$, the judgment process is as follows:

If $\Delta F^{\prime }\;<0$, indicating that the upcoming moving position is superior to the current position in terms of a weighted combination of safety and distance to the endpoint, the beetle moves to the next position. Otherwise, calculate
(18)$$P=\exp\left[-\frac{\Delta F^{\prime }}{\delta^{t}}\right]$$
where P is the probability of a move and r is the probability of an acceptable move. When P>r, although the position to be moved is worse than the current position, the beetle can still move to the next position with probability P. When P<r, in order to ensure the accuracy of the search, the beetle needs to re-search its next position to obtain the path that satisfies the highest safety and the shortest voyage.

### 2.6. Update Search Step Size and Sensing Length of Antennae

The search step size and the sensing length of antennae are crucial in determining the algorithm’s search speed and search accuracy, as well as the prerequisites for implementing safe obstacle avoidance and real-time search for waypoints. In this case, the larger the search step, the fewer positions need to be updated, so the search speed of the algorithm is faster. However, the search accuracy could be higher and the path obtained could be more precise and long. On the contrary, the smaller the search step, the more waypoints are planned, the shorter and smoother the path obtained, and the slower the algorithm’s search speed will be. Similarly, the longer sensing length of antennae, the larger the search range, but the lower the search accuracy. The shorter sensing length of antennae, the higher the accuracy of the search. Therefore, this paper improves the update method of the search step and the sensing length of antennae.

In solving other function optimization problems, how the BAS algorithm updates the step size is shown in Equation (19), where the step size attenuation factor α is generally set to 0.95 [16]. With each position update, the step size is reduced once by the attenuation factor. Therefore, the search precision gradually improves and converges to the optimal solution to the problem. When solving UAV path planning problems, the optimal value for the step size attenuation factor α is 0.99 [18], which can prevent the step size from decreasing too quickly to a small search step. The position updates of the beetle correspond to the search for waypoints for the UAV. The waypoints of the UAV are updated based on the step size. Significant differences in the search step size before and after may lead to unstable searching. The step size is too small in the later stage, which greatly reduces the search speed and may even prevent the beetle from reaching the target position. In response to this issue, this paper proposes a search strategy that interacts with fixed and variable step sizes, as shown in Equation (20):(19)$$\delta^{t+1}=\alpha \delta^{t}$$
(20)$$\delta^{t+1}=\left\{ \begin{array}{l} \delta^{0},D_{location}\notin O_{\lambda d_{\min}}\\ \begin{array}{l} -\frac{1}{2}d_{\min}+\frac{\delta^{0}-\;d_{\min}}{\lambda d_{\min}-3d_{\min}}d_{t,d}\;,3d_{\min}\leq d_{t,d}<\lambda d_{\min}\\ \frac{1}{4}d_{\min}+\frac{1}{4}d_{t,d}\;,\;d_{\min}\leq d_{t,d}<3d_{\min} \end{array}\}\;D_{location}\in O_{\lambda d_{\min}} \end{array}\right.$$
where, $\delta^{0}$ is the initial step size, $\delta^{t}$ and $\delta^{t+1}$ are the step sizes of the beetle at positions $X^{t}$ and $\;X^{t+1}$, $d_{\min}$ represents the minimum safe distance between the UAV and the obstacle, and $d_{t,d}$ is the distance between the UAV and the obstacle. $D_{location}$ represents the position of the obstacle, and $O_{\lambda d_{\min}}$ represents the area where danger exists. The range of the $O_{\lambda d_{\min}}$ area is a forward hemisphere with the current position of the beetle as the center and a radius of $\lambda d_{\min}$. $D_{location}\notin O_{\lambda d_{\min}}$ represents that there is no obstacle in the $O_{\lambda d_{\min}}$ area. Otherwise, $D_{location}\in O_{\lambda d_{\min}}$ represents the existence of an obstacle in the $O_{\lambda d_{\min}}$ area. According to the above equation, when the UAV is in the area without any obstacles, that is, far away from the obstacles, the current position of the UAV is relatively safe, and the UAV maintains its initial step size to search for the waypoint. When there are obstacles within range $O_{\lambda d_{\min}}$ of the UAV, which means the UAV is closer to the obstacle and there is a risk of flight. To ensure the flight safety of the UAV, it is necessary to reduce the search step size and search with a variable step size. The closer the UAV is to the obstacle, the more dangerous it is. Therefore, the search step size decreases as the distance between the UAV and the obstacle decreases, the closer the UAV is to an obstacle, the smaller the search step size, ensuring the safety of the UAV’s flight.

The original BAS algorithm updates the sensing length of the beetle’s antennae similarly to the update method of the search step size, as shown in Equation (21), which gradually decreases with the number of updates to the position. In the early stage of the search, the antennae use a longer sensing length to expand the search range, while in the later stage, a smaller sensing length is used to improve the search accuracy. However, considering the actual situation of UAV path planning, if the sensing length is too small, it will limit the search range and reduce the accuracy of the search. When solving the problem of UAV path planning, the updating method of the sensing length of antennae, like the updating method of the search step size, will reduce the search speed. Therefore, updating the sensing length of antennae in this paper is the same as the update method of the search step size mentioned above, which is determined based on the distance between the UAV and the obstacle, using alternating search with fixed sensing length and reduced sensing length. When no obstacles are detected in the area, exploring with a longer sensing length can accelerate the search speed. The closer the obstacle is, the more dangerous it becomes. Therefore, when there are obstacles in the area, the length of the antennae sensing needs to be reduced, based on the distance between the UAV and the obstacle. Therefore, the update of antennae sensing length is shown in Equation (22):(21)$$d^{t+1}=0.95d^{t}+0.01$$
(22)$$d^{t+1}=\sqrt{2}\delta^{t}$$
where, $d^{t}$ and $d^{t+1}$ are the step sizes of the beetle at positions $X^{t}$ and $\;X^{t+1}$.

The Beetle Antennae Search Simulated Annealing algorithm is given in Algorithm 1.

**Algorithm 1** Improvement of the Beetle Antennae Search Simulated Annealing Algorithm**Input:**X0, XT, δ0, d0, *M*;**Output:** flight path;  Initializing the search direction;
  **Repeat**
    Update the left and right antennaes and the next position of the beetle $X^{t+1}$;    jump_point1:      Generate the neighborhood solution of $X^{t+1}$;      **for** *m* = 1:*M* do         **if** ΔF<0 then         $X^{t+1}=X_{m}^{t+1}$;        **end if**
      **end for**
      Get the next best position;      **if** $\Delta F^{\prime }<0$ then         Move to the next step;        **else if** r<P then        Move to the next step;        **else** goto jump_point1;
        **end if**

      **end if**
      Update the search step size and the sensing length of antennae;
  **Until**
$X^{t+1}=X^{T}$
return flight path

The initial search direction of the BAS algorithm is randomly generated, leading to a relatively haphazard search with low accuracy. The concentration difference between its two antennae determines the beetle’s position update, resulting in high limitations and low accuracy, making it susceptible to getting trapped in local optimal positions. Both search step size and antennae sensing length gradually decrease, which makes the search speed fast and low accuracy in the early stage, and slow and high accuracy in the later stage. Addressing the issues with the BAS algorithm, the SABAS algorithm sets the search direction as the direction from the starting point to the destination, to enhance search accuracy. Moreover, to prevent falling into local optima, the process of updating positions incorporates the SA algorithm, which can introduce neighboring solutions to expand the search range. The Metropolis criterion of the SA algorithm is used to determine whether the updated position meets the conditions for movement, avoiding the beetle from advancing mindlessly. In areas far from obstacles, a larger step size and antennae sensing length are used for search. In areas close to obstacles, a gradually decreasing variable step size and antennae sensing length are employed for search. This method of using variable step size and antennae sensing length for search ensures accuracy and speed in the search process. Therefore, the SABAS algorithm can meet logistics UAVs’ requirements for real-time path planning in low-altitude urban environments. 

## 3. Simulation Experiments and Result Analysis

The experimental planning area in this paper is a three-dimensional space of 1000 m × 1000 m × 100 m, where each unit length along the axis z is 1 m, and each unit length along the axes x and y is 20 m. The starting and ending coordinates of the planned path for the UAV are (0, 0, 65) and (1000, 1000, 85), respectively. The minimum safety distance $d_{\min}$ between the UAV and obstacles is set to 10 m, the dangerous area $\lambda d_{\min}$ is $5d_{\min}$, the initial step size $\delta^{0}$ is 20 m, the initially sensing length of the antennae is $\sqrt{2}*20$ m, the r value is 0.8, the maximum flight altitude $High_{\max}$ is set to 120 m, the minimum flight altitude $High_{\min}$ is set to 30 m, the maximum turning angle $\beta_{\max}$ and the maximum pitch angle are $\mu_{\max}$ both set to $\frac{\pi }{2}$, $c_{1}$, $c_{2}$, and $c_{3}$ are taken as 0.3, 0.3, and 0.4 respectively. To avoid algorithm limitations specific to certain scenarios and to better test the algorithm’s general applicability, this study has established three complex urban low-altitude environments: obstacle-sparse, obstacle-moderate, and obstacle-dense. These three scenarios differ only in the density and distribution of obstacles; all other parameters are kept consistent. To thoroughly evaluate the performance of the proposed algorithm, the original BAS algorithm is used as a control experiment. The simulation results are as follows.

Figure 1 and Figure 2 show the three-dimensional diagram and top view of the paths planned by the SABAS and BAS algorithms in an obstacle-sparse environment. Figure 3 and Figure 4 show the three-dimensional diagram and top view of the paths planned by the two algorithms in an obstacle- moderate environment. Figure 5 and Figure 6 show the three-dimensional diagram and top view of the paths planned by the two algorithms in an obstacle- dense environment. The solid red line is the flight path planned by the SABAS algorithm, and the solid black line is the flight path planned by the BAS algorithm. As shown in the figure, both algorithms can plan an effective flight path, and the path obtained by the SABAS algorithm has fewer sharp turns and relatively gentle turns. That is because the SABAS algorithm expands the search range and does not limit the position of the next waypoint. In addition, since the SABAS algorithm considers the safe distance between UAV and obstacles in the fitness function, the distance between each waypoint of the path and obstacles is within the safe range. In contrast, the BAS algorithm relies only on the difference between the left and right antennae concentrations to determine the position. Therefore, the path is more tortuous, increasing the UAV energy consumption and threatening flight safety. In order to make the flight path smoother, more studies will smooth the path after algorithm planning. But this method is only suitable for offline path planning and vastly reduces planning efficiency.

Table 1 shows the data of the path results planned by the SABAS algorithm and BAS algorithm for the three scenarios, respectively. The difference between the flight waypoints planned by the two algorithms is the most noticeable. Among them, the BAS algorithm searched 1.7 times more waypoints than the SABAS algorithm in the obstacle-sparse environment, 1.35 times more in the obstacle-moderate environment, and 1.23 times more in the obstacle-dense environment. The SABAS algorithm searched 127 waypoints in the obstacle-dense environment, 26 more than in the obstacle-moderate environment and 41 more than in the obstacle-sparse environment. The BAS algorithm searched 157 waypoints in the obstacle-dense environment, 5 more than in the obstacle-moderate and 10 more than in the obstacle-sparse environment. It can be concluded that the waypoints planned by the SABAS algorithm have advantages over those planned by the BAS algorithm in all three scenarios. Among them, the number of waypoints that need to be searched has a more obvious advantage in obstacle-sparse environments. That is because the search step size of the BAS algorithm gradually decreases, resulting in smaller step sizes for subsequent searches. Therefore, in all three scenarios, there are more waypoints to search for and the difference is relatively small. In contrast, the search step size of the SABAS algorithm is updated based on the distance between the UAV and obstacles, searching for waypoints with a longer step size in areas far from obstacles, resulting in fewer waypoints that need to be searched compared to the BAS algorithm. Meanwhile, the search step needs to be reduced close to the obstacle, so the number of waypoints to be searched by the SABAS algorithm in the obstacle-dense environment increases more than that in the obstacle-sparse environment.

In addition, in the obstacle-sparse environment, the path length planned by the SABAS algorithm is 33 m less than that planned by the BAS algorithm. In the obstacle-moderate environment, the path length planned by the SABAS algorithm is 41 m less than that planned by the BAS algorithm. In the obstacle-dense environment, the path length planned by the SABAS algorithm is 46 m less than that planned by the BAS algorithm. As a result, the paths obtained by the SABAS algorithm are significantly better than the BAS algorithm in terms of path length in both environments, and the advantage is more evident in the obstacle-dense environment. That is because the BAS algorithm only updates the position based on the concentration of left and right antennae, resulting in relatively random and low-accuracy searches. Furthermore, moving with a larger step size in the front leads to inaccurate search and deviation from the optimal route, which increases the path length and cannot better avoid obstacles; moving in smaller steps later slows down the search speed. The SABAS algorithm sets the initial search direction as the direction from the starting point to the endpoint, ensuring that the UAV is searching toward the endpoint, improving the accuracy of the search. The SA algorithm is combined with the BAS algorithm to update the waypoints, expanding the search range of each waypoint to avoid falling into the local optimal waypoints in updating the waypoint process. Moreover, the effectiveness of the next waypoint is evaluated based on Metropolis criteria, effectively avoiding the blind advance of UAVs. The search step size and the sensing length of antennae are updated according to the distance between the beetle and the obstacle, and the speed of search is improved by moving with a fixed step size within a safe range from the obstacle, while moving with a smaller step size in the place of the obstacle improves the real-time performance of path planning and the accuracy of the search. It can be obtained that, compared with the original algorithm, the SABAS algorithm is more adaptable to environments with different obstacle complexities, and the more prominent the advantage is in complex urban environments.

From the above analysis, the path planned by the algorithm in this paper is smoother. It has advantages over the BAS algorithm in terms of path length and waypoints, which can meet the needs of logistics UAVs for real-time path planning at low altitudes in cities.

## 4. Conclusions

A modified path planning method SABAS algorithm is proposed, which combines the BAS with SA, to address the issue of poor real-time performance in path planning algorithms for urban low-altitude logistics UAVs. This algorithm is designed with a fitness function, which considers safety obstacle avoidance and the shortest path requirement, effectively meeting the demands of real-time path planning. The method of determining the initial direction in the BAS algorithm is improved, enabling the search step length and antennae sensing length to be updated in real time based on the distance between the UAV and obstacles. This enhancement significantly enhances the safety and real-time performance of the search process. Integrating the SA algorithm into the BAS algorithm for waypoint updates, the process is prevented from becoming trapped in the local optimal waypoint. Moreover, utilizing the Metropolis criterion to evaluate the next waypoint, the accuracy of the search is improved. From the simulation results, in the three complex urban environments with sparse, medium, and dense obstacles, the path length planned by the SABAS algorithm is reduced by 33 m, 41 m, and 46 m than that planned by the BAS algorithm, respectively, and the waypoints searched by the SABAS algorithm are reduced by 61, 40 and 30 more than that searched by the BAS algorithm, respectively. Comparing the simulation results, it is evident that the SABAS algorithm exhibits greater versatility in low-altitude urban environments. It is capable of generating safe obstacle avoidance and smooth paths. The planned paths of the SABAS algorithm outperform the BAS algorithm in terms of both waypoints and path length. As a result, the SABAS algorithm can effectively achieve real-time path planning for urban low-altitude logistics UAVs.

## Figures and Tables

**Figure 1 sensors-23-07472-f001:**
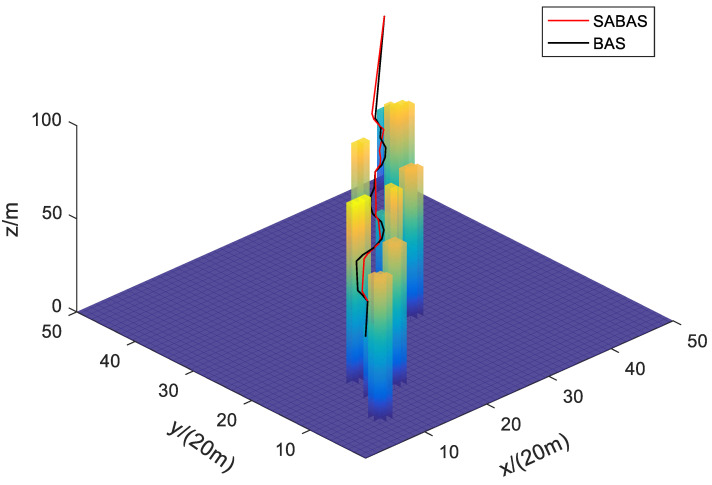
Three-dimensional map of path in the obstacle-sparse environment.

**Figure 2 sensors-23-07472-f002:**
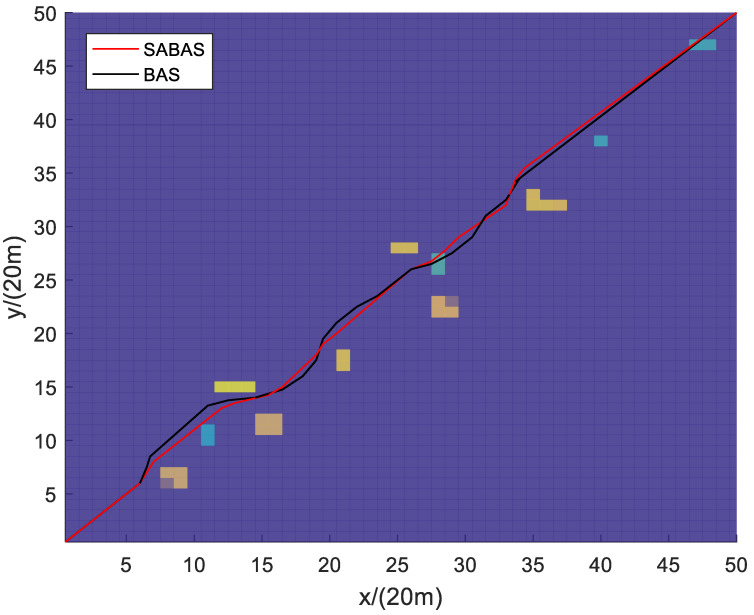
Top view of the path in the obstacle-sparse environment.

**Figure 3 sensors-23-07472-f003:**
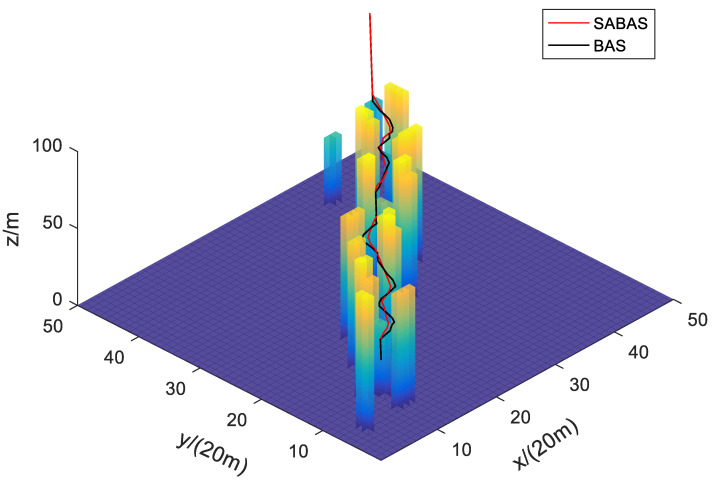
Three-dimensional map of path in the obstacle-moderate environment.

**Figure 4 sensors-23-07472-f004:**
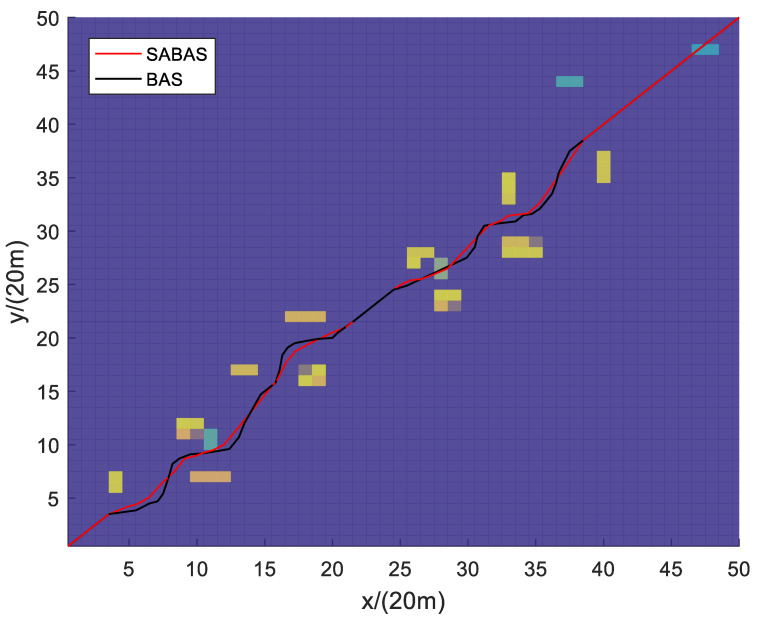
Top view of the path in the obstacle-moderate environment.

**Figure 5 sensors-23-07472-f005:**
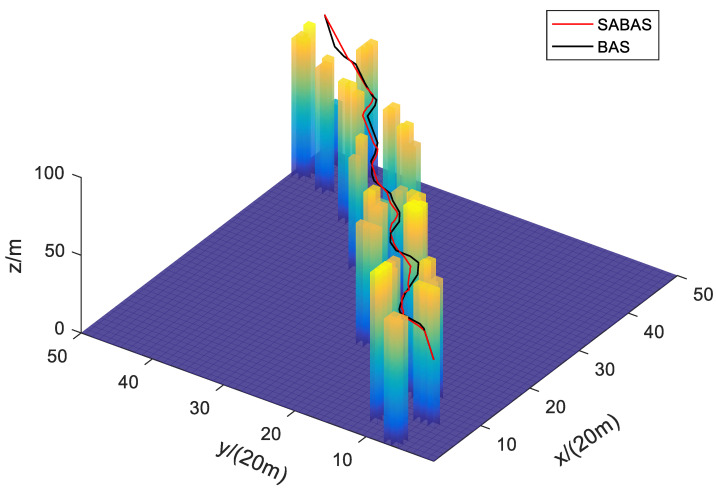
Three-dimensional map of path planning in the obstacle-dense environment.

**Figure 6 sensors-23-07472-f006:**
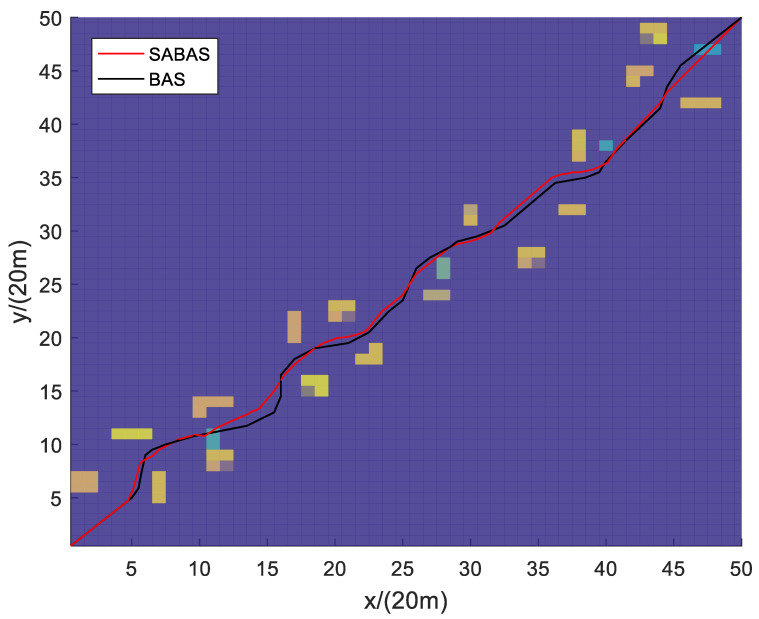
Top view of path planning in the obstacle-dense environment.

**Table 1 sensors-23-07472-t001:** Path Planning Result Data Table.

Environment	Path Length/m		Waypoints	
	SABAS	BAS	SABAS	BAS
obstacle-sparse	1489	1522	86	147
obstacle-moderate	1506	1547	112	152
obstacle-dense	1523	1569	127	157

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
