# Peer review of "A Real-Time Path Planning Method for Urban Low-Altitude Logistics UAVs"

_sensors, 2023, doi:10.3390/s23177472_

Round 1
Reviewer 1 Report
In this work, an modified Beetle Antennae Search based on simulated annealing algorithm is proposed to improve the search accuracy and real-time performance of path planning. Firstly, by designing the fitness function in Beetle algorithm, the search step size and the sensing length of antennae are changed in real time, which improves the real-time performance of the path planning algorithm. Secondly, in the study of optimal path planning, many algorithms usually face the limitation of local optimal. The author applies the annealing algorithm to Beetle algorithm to track the point update, which effectively avoids the limitation of local optimal.
I propose the following:
Comment 1:Adding symbol descriptions, such as the initial step δ0, should appear where the symbol first appears to increase the readability of the article.
Comment 2: In the "introduction" part, improve the literature generation, highlight the contribution of the article, especially the advantages of the introduction of annealing algorithm in solving the local optimal.
Comment 3: It is suggeted to struscture your contributions more carefully and focus on the insights generated in your paper.
- Most paragraphs of the article have good logic and fluency. Only minor changes are required.
Author Response
Response to Reviewer 1 Comments
Thank you very much for your careful review and affirmative comments on our manuscript. According to your opinion, we have carefully revised the manuscript. Changes in the revised manuscript are clearly marked by red color.
Point 1: Adding symbol descriptions, such as the initial step δ0, should appear where the symbol first appears to increase the readability of the article.
Response 1: Thanks very much for your suggestions. According to your advice, we carefully reviewed the description of each symbol in the paper, ensuring that descriptions were added where symbols first appeared, thus ensuring the article's readability.
Point 2: In the "introduction" part, improve the literature generation, highlight the contribution of the article, especially the advantages of the introduction of annealing algorithm in solving the local optimal.
Response 2: Thanks very much for your suggestions. According to your advice, we have revised the introduction and added relevant references to emphasize the contributions of the paper. In particular, we have included references [24], [25], and [26] to highlight the advantages of the simulated annealing algorithm in addressing the issue of other algorithms falling into local optima. The main structure of the introduction is described as follows: Firstly, we describe the current situation and practical significance of logistics UAV path planning research, and point out the current problems of urban low-altitude logistics UAV path planning. Then we point out the methods and problems of path planning, and propose the solutions to the problems.
Point 3: It is suggeted to struscture your contributions more carefully and focus on the insights generated in your paper.
Response 3: Thanks very much for your suggestions. According to your advice, we have revised the summary of the paper's contributions in the introduction. At the same time, we have focused on the insights generated within the paper, made changes to the content of the conclusion section, and highlighted the paper's contributions in terms of enhancing safety obstacle avoidance, reducing waypoint searching, and shortening flight path length. The primary contributions and conclusions of this paper are as follows:
Contributions:
(1) Considering the constraints during UAV flight, design a fitness function that satisfies safe obstacle avoidance, shortest path, and real-time planning requirements.
(2) Improving the BAS algorithm. To enhance search safety and real-time perfor-mance, the search step size and antennal sensing length are dynamically updated based on the distance between the UAV and obstacles. When the UAV is far from obstacles, larger step sizes and antennal sensing lengths are used for searching. In comparison, smaller step sizes and antennal sensing lengths are employed when the UAV is closer to obstacles.
(3) Applying the SA algorithm to the position updating of the BAS algorithm. The search range is expanded by generating neighboring solutions for the next position, preventing the position update process from falling into local optima. Additionally, assessing whether the next position meets the movement criteria based on the Metropolis criterion, to enhances the accuracy of the search.
(4) Comparing with the BAS algorithm in three urban low-altitude scenarios, where the only difference among these scenarios is the obstacle density and distribution. Test the algorithm's performance in terms of safe obstacle avoidance, number of waypoints, and flight path length.
Conclusions:
A modified path planning method SABAS algorithm is proposed, which combines the BAS with SA, to address the issue of poor real-time performance in path planning algorithms for urban low-altitude logistics UAVs. This algorithm is designed with a fitness function, which considers safety obstacle avoidance and the shortest path requirement, effectively meeting the demands of real-time path planning. The method of determining the initial direction in the BAS algorithm is improved. Updating search step length and antennae sensing length in real-time based on the distance between the UAV and obstacles. This enhancement significantly enhances the safety and real-time performance of the search process. Integrating the SA algorithm into the BAS algorithm for waypoint updates, the process is prevented from becoming trapped in local optimal waypoint. Moreover, utilizing the Metropolis criterion to evaluate the next waypoint, the accuracy of the search is improved. From the simulation results, in the three complex urban environments with sparse, medium, and dense obstacles, the path length planned by the SABAS algorithm is reduced by 33m, 41m, and 46m than that planned by the BAS algorithm, respectively, and the waypoints searched by the SABAS algorithm are reduced by 61, 40 and 30 than that searched by the BAS algorithm, respectively. Comparing the simulation results, it is evident that the SABAS algorithm exhibits greater versatility in low-altitude urban environments. It is capable of generating safe obstacle avoidance and smooth paths. The planned paths of the SABAS algorithm outperform the BAS algorithm in terms of both waypoints and path length. As a result, the SABAS algorithm can effectively achieve real-time path planning for urban low-altitude logistics UAVs.

Reviewer 2 Report
A path planning approach combining modified Beetle Antennae Search with simulated annealing algorithm is proposed in this paper. However, this paper must be revised completely, before it can be detailed reviewed.
First, the reason for applying the proposed method is amphibolous. Why the Beetle Antennae Search is proper in this issue? The advantage and feasibility are not clearly expressed.
Then, the simulatiuon environment is too simple. The barriers are all located on the diagonal line. This problem can be solved by too many method. I can not see the necessity and novelty of the proposed method in this condition.
The article format is also not rigorous. For example, all an equation, it exists both "Where" and "where". Which one is correct?
The expression should be improved.
Author Response
Response to Reviewer 2 Comments
Thank you very much for your careful review and affirmative comments on our manuscript. According to your opinion, we have carefully revised the manuscript. Changes in the revised manuscript are clearly marked by red color.
Point 1: The reason for applying the proposed method is amphibolous. Why the Beetle Antennae Search is proper in this issue? The advantage and feasibility are not clearly expressed.
Response 1: Thanks very much for your suggestions. According to your advice, we have revised the introduction to highlight the reasons for the methods proposed in this paper: Currently, the real-time performance of path planning methods for logistics UAVs is poor, with many relying on offline planning approaches that suffer from issues such as lengthy planning times, poor autonomy, and weak adaptability to the environment. Additionally, we have supplemented the advantages and feasibility of the Beetle Antennae Search Algorithm. As one of the most typical search algorithms, the Beetle Antennae Search Algorithm requires only a single individual, is not constrained by initial conditions, and has fewer constraints.. This makes it easier to integrate with other algorithms. Additionally, it possesses lower complexity and stronger search capabilities. Therefore, it holds a distinct advantage in addressing real-time path planning challenges for UAVs.
Point 2: The simulatiuon environment is too simple. The barriers are all located on the diagonal line. This problem can be solved by too many method. I can not see the necessity and novelty of the proposed method in this condition.
Response 2: Thanks very much for your suggestions. According to your advice, we have added a scenario description to the first paragraph of the simulation results analysis. In a real-world environment, the density of obstacles is random, and their distribution is irregular. To avoid the algorithm's limitations imposed by specific scenarios and to better assess its general applicability, we have set up three scenarios with different densities and distributions of obstacles. The obstacles in the simulated scenario do appear to be positioned along the diagonal, making the simulated scenario look simple.
The reasons for setting up the simulated scenarios in this manner are as follows:
Firstly, the fitness function designed in this paper is composed of danger cost, altitude cost, and distance cost to the endpoint. The position update of the beetle corresponds to the waypoint update of the UAV. Therefore, the UAV path planning process in this paper is essentially the UAV searching for relatively safe points and closer to the endpoint, and these sequentially connecting these waypoints will be the UAV's path. Secondly, according to the algorithm's search method, the initial direction is is from the starting point towards the endpoint, and obstacles at a certain distance from the UAV's current waypoint have no effect on the update of the UAV's next waypoint. It can be concluded that in a scenario without obstacles, the path should be a direct line between the starting and ending points. In the presence of obstacles, the path would move closer to the direction from the starting point to the ending point after avoiding the obstacles. In previous experiments, besides the obstacles shown in these simulated images, additional obstacles were also included. However, according to the fitness function and search method of the algorithm in this paper, only the obstacles in the simulation graph affect the formation of paths in the path generation process. Therefore, to facilitate a clearer and more intuitive observation of the simulation results, we have removed obstacles that do not affect the path. In many studies, it is common to retain only the obstacles that have an impact on the path in order to better showcase or compare the obtained paths.
The necessity and novelty of the proposed method in this paper lie in the following aspects:
The algorithm's designed fitness function, as expressed in Equation (1), considers the safe distance between the UAV and obstacles. As a result, each waypoint of the path maintains a safe distance from obstacles. Setting the initial search direction of the algorithm from the starting point to the destination ensures that the UAV progresses toward the endpoint, enhancing the accuracy of the search and preventing unnecessary detours. The SA algorithm is applied to the BAS algorithm for waypoint updating to expand the search range of each waypoint, thus avoiding the process of updating waypoints from falling into local optimal waypoints. And combined with the Metropolis criterion to evaluate the effectiveness of the next waypoint, effectively avoiding the UAV blindly moving forward. The search step size and antennae sensing length are updated according to the distance between the beetle and the obstacle, moving with a larger step size far from the obstacle improves the speed of the search, while moving with a smaller step size near the obstacle improves the real-time of the path planning and the accuracy of the search. So, it is possible to obtain paths with fewer waypoints, shorter path length and smoother paths. As a result, the algorithm is capable of meeting the real-time path planning requirements of logistics UAVs in low-altitude urban areas.
Point 3: The article format is also not rigorous. For example, all an equation, it exists both "Where" and "where". Which one is correct?:
Response 3: Thanks very much for your suggestions. According to your advice, we have thoroughly reviewed the format of the paper, corrected any non-standard formatting, and ensured consistent wording.

Reviewer 3 Report
The paper is easy to read. The title and abstract are coherent with the text. BAS and SABAS algorithms were investigated in a simulation. There are some typos still in the text. For instance: "The experimental planning area in this article is a 3D space of 1000 * 1000 * 100. The 346 logistics UAV selected is a Multi-rotor UAV. The starting and ending coordinates of the 347 planned paths for the UAV are (0,0,65) and (10001000,85), respectively." - I believe the coordinates are bad!
The comparison of BAS and SABAS algorithms is missing. Please extend it.
What does z/m, y/(20m) and x/(20m) means on figures?
How do these axes are connected to the 1000*1000*100 pixels?
Could these papers be cited?
Bagdi, Z., Csámer, L., & Bakó, G. (2023). The green light for air transport: sustainable aviation at present. Cognitive Sustainability, 2(2). DOI: https://doi.org/10.55343/cogsust.55
Bauer, P., Bokor, J. (2008) “LQ Servo control design with Kalman filter for a quadrotor UAV”, Periodica Polytechnica Transportation Engineering, 36(1-2), pp. 9–14. https://doi.org/10.3311/pp.tr.2008-1-2.02
Author Response
Response to Reviewer 3 Comments
Thank you very much for your careful review and affirmative comments on our manuscript. According to your opinion, we have carefully revised the manuscript. Changes in the revised manuscript are clearly marked by red color.
Point 1: The paper is easy to read. The title and abstract are coherent with the text. BAS and SABAS algorithms were investigated in a simulation. There are some typos still in the text. For instance: "The experimental planning area in this article is a 3D space of 1000 * 1000 * 100. The 346 logistics UAV selected is a Multi-rotor UAV. The starting and ending coordinates of the 347 planned paths for the UAV are (0,0,65) and (10001000,85), respectively." - I believe the coordinates are bad!
Response 1: Thanks very much for your suggestions. According to your advice, we have corrected the erroneous coordinates (10001000,85) to the correct coordinates (1000,1000,85) within the paper. Additionally, we have thoroughly reviewed the entire paper and rectified other errors present in the paper.
Point 2: The comparison of BAS and SABAS algorithms is missing. Please extend it.
Response 2: Thanks very much for your suggestions. According to your advice, in the final section of Chapter 2, we have added a more detailed comparison between the BAS and SABAS algorithms. In addition, the principles of the algorithms are combined in analyzing the experimental results, and the paths planned by the two algorithms are compared in detail. It is shown that the SABAS algorithm is more universal to low-altitude urban environments, can generate safe obstacle avoidance and smooth paths, and the paths are better than the BAS algorithm in terms of waypoints and range, which can realize the real-time planning of paths for urban low-altitude logistics UAVs.
The description of the algorithm comparison is as follows:
The initial search direction of the BAS algorithm is randomly generated, leading to a relatively haphazard search with low accuracy. The concentration difference between its two antennae determines the beetle's position update, resulting in high limitations and low accuracy, making it susceptible to getting trapped in local optimal positions. Both search step size and antennae sensing length gradually decrease, which makes the search speed fast and low accuracy in the early stage, and slow and high accuracy in the later stage. Addressing the issues with the BAS algorithm, the SABAS algorithm sets the search direction as the direction from the starting point to the destination, to enhance search accuracy. Moreover, to prevent falling into local optima, the process of updating positions incorporates the SA algorithm, which can introduce neighboring solutions to expand the search range. The Metropolis criterion of the SA algorithm is used to determine whether the updated position meets the conditions for movement, avoiding the beetle from advancing mindlessly. In areas far from obstacles, a larger step size and antennae sensing length are used for search. In areas close to obstacles, a gradually decreasing variable step size and antennae sensing length are employed for search. This method of using variable step size and antennae sensing length for search ensures accuracy and speed in the search process. Therefore, the SABAS algorithm can meet logistics UAVs' requirements for real-time path planning in low-altitude urban environments.
Point 3: What does z/m, y/(20m) and x/(20m) means on figures?
Response 3: Thanks very much for your suggestions. According to your advice, we have added explanations to the first paragraph of the simulation results analysis regarding the coordinate axes. In this context, z/m indicates that each unit length on the z-axis represents 1 meter, while y/(20m) and x/(20m) indicate that each unit length on the y-axis and x-axis corresponds to 20 meters.
Point 4: How do these axes are connected to the 1000*1000*100 pixels?
Response 4: Thanks very much for your suggestions. According to your advice, we have added explanations to the first paragraph of the simulation results analysis, describing the coordinate axes and the planning area. Due to improper writing, there is an error in the description of the 1000 * 1000 * 100 region for path planning. In this paper, 1000 * 1000 * 100 is a three-dimensional space of 1000m * 1000m * 100m. In this context, z/m indicates that each unit length on the z-axis represents 1 meter, while y/(20m) and x/(20m) indicate that each unit length on the y-axis and x-axis corresponds to 20 meters. The z-axis has 100 units, while the x and y-axes have 50 units, corresponding to a three-dimensional space of 1000m * 1000m * 100m.
Point 5: Could these papers be cited?
Bagdi, Z., Csámer, L., & Bakó, G. (2023). The green light for air transport: sustainable aviation at present. Cognitive Sustainability, 2(2). DOI: https://doi.org/10.55343/cogsust.55
Bauer, P., Bokor, J. (2008) “LQ Servo control design with Kalman filter for a quadrotor UAV”, Periodica Polytechnica Transportation Engineering, 36(1-2), pp. 9–14. https://doi.org/10.3311/pp.tr.2008-1-2.02
Response 5: Thank you very much for recommending these two papers for our study. After conducting our research, we found that the quality of these two papers is exceptionally high, which has proven immensely beneficial to our research. The contributions of these two papers have lent greater persuasiveness to our work. References [5] and [9] are citations from these two papers.
